# Optimal three-part tariff pricing and marketing strategies for consumer overconfidence

Silvia Merdikawati[1], Shi-Woei Lin[1,2]*, Ruey Huei Yeh[1]

1 Department of Industrial Management, National Taiwan University of Science and Technology, Taipei City, Taiwan, 2 Artificial Intelligence for Operations Management Research Center, National Taiwan University of Science and Technology, Taipei, Taiwan

* shiwoei@mail.ntust.edu.tw

**Data Availability Statement:** All relevant data are within the manuscript and its Supporting Information files.

**Funding:** S.W. Lin Grant Number: MOST110-2621-M-011-001 Grant Number: MOST106-2410-H-

## Abstract

Experts in traditional pricing literature often assume that consumers are rational in their purchasing decisions, and tend to ignore the effect of psychological behavior. One of the fundamental irrational psychological behaviors is overconfidence, which refers to overestimation. The benefits of three-part tariffs under demand uncertainty caused by overconfident consumers have been demonstrated in various forms. In the present study, we investigate the design of a three-part tariff pricing plan when consumers are overconfident in the sense of overestimating their amount of service usage or consumption. An analytical model and a numerical analysis are carried out using a genetic algorithm approach to determine the optimal three-part tariff and to compare other tariff structures (e.g., two-part tariff and menu of three-part tariff) to investigate the critical factors influencing their performance. With overconfidence in consumers, service providers need clear direction on how to optimally implement a three-part pricing strategy. Specifically, we observe the effect of overconfidence on the profitability of service providers, and their other performance in general. Thus, service providers are given clear guidance on when to offer three-part tariffs and alternative structures to overconfident consumers and push marketing plans to increase their profits and market share.

## Introduction

Service providers typically offer a three-part tariff that includes a fixed cost, unit price, and a service provider–provided unit allowance. Wireless phone services, Internet service, car leasing, online music download, and credit cards are all instances of three-part tariffs. In the telecommunications business, service providers often charge customers a fixed monthly subscription, number of free minutes, and a usage fee when the free amount is exceeded. Traditional pricing literature frequently assumes that consumers are rational in their purchase decisions, ignoring the effect of psychological behavior on usage divergence from reality. However, consumer views frequently and significantly stray from rational expectations in a systematic manner [1, 2].

011-004-MY3 Funder: Ministry of Science and Technology, Taiwan http://www.most.gov.tw The funders had no role in study design, data collection and analysis, decision to publish, or preparation of the manuscript.

**Competing interests:** The authors have declared that no competing interests exist.

In terms of psychological behaviour, overconfidence is one of the most prominent problems in decision making (e.g., [3–5]). From the perspective of consumers' behaviour, overconfidence may lead to an overly optimistic view of their ability being greater than their objective (original) consumption (e.g., [1, 5–9]). For instance, an overconfident consumer may overestimate his/her likelihood of consuming his/her Internet access, with a large gigabyte capacity. Service providers thus have an incentive to complicate their pricing structures (e.g., provide several pricing plans) to induce different customers to reveal their preferences and increase the consumption on demanded service [8, 10, 11]. By creating complicated pricing structures, service providers can better understand the consumption patterns and behaviours of overconfident consumers and understand how and when different tariff structures should be adopted [1]. Hence, examining the impact of overconfident consumers has important implications for pricing design and service providers' profits.

According to Li et al. [12], when consumer overconfidence is high, the company's profit improves. If customers are overconfident, then the retailer is more inclined to perform regular advance selling. The retailer can use marketing strategies such as advertising and promotion to boost consumers' overconfidence, resulting in increased profits. Li and his colleagues examined how retailers choose to charge greater prices in the follow-up sales period when consumers are overconfident. Ren et al. [13] discovered that the order bias is linear at an overwhelming degree of confidence and grows with the variance of demand distribution in another investigation. Indeed, research in behavioral theory emphasizes that consumer misprediction allows service providers to increase profits when entering the market [5, 14, 15].

In the service industry, consumer and service provider decisions are both influenced by such overconfidence. Several consumers, for example, have a clear understanding of their demand distribution while others have misconceptions, and in particular, overestimate their ability to extract all usage or demand from the service. Grubb [1] reported on the benefits of three-part tariffs under demand uncertainty caused by customer overconfidence; he recommended service providers to offer a three-part tariff pricing plan, particularly for new customers who are overconfident about their amount of consumption when they first enter the market. The three-part tariff is selected because consumers are confident that all of the surplus are to be consumed, but customers actually consume less than they believe. The consumers overvalue their usage by an extra $1, which becomes an extra profit for the company. Thus, consumers that place a high value on their consumption commonly benefit service providers.

The study on the optimal three-part tariff pricing has been merely investigated analytically in a few research, and the literature on three-part tariff pricing is mostly empirical or numerical (e.g., [16–20]). Specifically, two studies conducted analytical analyses on the optimal three-part tariff optimization problem, namely, Grubb [21] and Fibich et al. [22]. For overconfident consumers, Grubb [21] investigated the optimal three-part tariff pricing by assuming that each customer has an estimated demand and actual demand and chooses a plan based on his/her estimated demand. The author demonstrated that when customers are overconfident, companies can make more money. Fibich et al. [22] considered a symmetric information between companies and consumers as well as a theoretical model in which a service provider offers services to risk-averse and rational consumers. The main contribution of our study is different from that of Grubb [21] and Fibich et al. [22], which focused only on the optimal three-part tariff. In our study, we also include cognitive bias in the model formulation and consider investigating the three-part tariff pricing menu.

This study models such overconfidence in demand as consumers overestimate their amount of service usage or consumption. The design of three-part tariff pricing plans when customers are overconfident is investigated by formulating an analytical model and carrying out a numerical analysis using genetic algorithm approaches to determine the optimal values

and to compare other structures (e.g., two-part tariff and menu of three-part tariffs). In addition, we intentionally change the parameters or use different parameter values to explore the properties of the optimal three-part tariff. In the presence of overconfidence, service provider requires explicit directions on how to implement the three-part tariff pricing menu strategy. Specifically, we examine the effect of overconfidence on the profits of service providers and attempt to provide clear advice on when to offer the three-part tariff menu to overconfident customers. Furthermore, in our model, we consider the presence of asymmetric information between service providers and individual consumers, because service providers know the distribution of all the consumers but do not know each consumer's preference. However, individual consumers have superior information, because they know their demand function.

The remainder of our article is organized as follows: First, we present the analytical model of the tariff optimization problem. Following this, we formulate the mathematical model of the three-part tariff pricing with consumer overconfidence. For the results part, we begin with a comparative statics analysis, followed by a numerical analysis and an experimental study on the different menus of three-part tariffs to compare the profitability of various tariff structures. Finally, we summarize our findings in the concluding section.

## Analytical model: Optimal pricing of single three-part tariff with consumer overconfidence

First, we model the consumer overconfidence in terms of their beliefs preference and usage behavior based on rational consumer demand. Second, we drive the profit function for the service provider when implementing a three-part tariff pricing scheme that comprises a fixed fee (T), unit price ($p_0$), and a usage allowance ($q_0$) under a fixed fee.

We suppose that among several consumers, each is distinguished by the preference parameter O (the preference of the original consumers), with a range of 0–1, and a high O value suggests high preference for the service. In other words, if $O_2 > O_1$, then the $O_2$ consumers' demand curve is above or to the right of that of the $O_1$ consumers. We expect consumer utility to increase linearly with O and the two demand curves to not intersect as a result. The probability density of the original consumer type (O) is uniformly distributed, which represents the variability or diversity between the consumers' preferences; therefore, its function is f(O) = 1 for $0 \leq O \leq 1$. In our approach, we consider a situation in which asymmetric information exists between a service provider and individual consumers. A monopolist service provider who is maximizing its profits is aware of the f(O) distribution of all the consumers but uncertain about which consumers he/she is facing, because the preference of the consumers is randomly generated from the distribution. By contrast, the consumers will have superior information, because they are aware of their own demand functions.

Following Tirole [23], we build a simple consumer demand function, where demand is assumed to be a linear function (with a non-negative intercept and constant negative slope) of price and independent of time. This function is extensively used in economics and in the operations management literature (see [16, 17, 20, 24–27]). In particular, the linear demand function can be justified in terms of utility theory. In other words, a demand function can be derived from a utility function together with a budget constraint. Thus, with reasonable assumptions of the consumer's behaviour in real world practices that under a given budget they will maximize the utility function that is quadratic in the usage consumption, the first order conditions for the consumer problem do result in linear demand when the utility function is strictly concave in the quantities consumed. Since this utility function is relatively flexible and can be applied to many real-world cases, we can justify the appropriateness of the linear demand function in terms of utility theory and considerably simplify the analytical

model in this study. The linear demand function is defined by Eq (1):

$$q = O - \frac{1}{\lambda}p, \tag{1}$$

where q is the quantity demanded by the consumers; p is the price of the goods or services; the constant $\lambda$ (or $\frac{1}{\lambda}$) is the slope of the demand curve, which shows the sensitivity of the quantity demanded for the product or service price; and the intercept O represents the original consumer type, with varying preferences and used to formulate the aggregate effect of the non-price factors on the quantity demanded. For a given price, the quantity demanded is high when preference O is high.

In the following section, we use the basic model of Jang and Kwon [28], first assuming that the density function of the original consumer type obeys a uniform distribution, with a probability density function f(O) = 1. We assume that the cost function C(·) has low constant marginal costs, because variable costs in the telecommunications industry are typically low. Then, C(Q) = mc. Q, where Q is the total consumption, and mc is the constant marginal costs.

In this study, we define consumers' overconfidence using the definition in Dellavigna and Malmendier [10], that is, the measure of consumers' overconfidence is given by the difference between the estimated (forecasted) and original consumption. In this situation, the overconfident consumers overestimate their future demand and ability to take advantage of their consumption of the demanded service as well as the units consumed. From the service provider's perspective, usage overestimation is equivalent to an upward shift in demand. This condition is illustrated in Fig 1, given that among the overconfident consumers, the demand curve shifts upward by (r), which is the amount that the consumers overestimate their unit consumption from the original consumer preference (O). Thus, we use the following expression to explain the demand of the overconfident consumers:

$$q = (O + r) - \frac{1}{\lambda}p. \tag{2}$$

The notation O+r is a special way to define the consumers' overconfidence, which we consider as a type of overestimation and means that after O+r, the consumers will become overconfident. Notation O denotes a preference parameter of the original consumer type. The constant r is a consumer's level of overconfidence that he/she will consume a number of units. We assume that the overconfidence level (r) is the same for all the consumers. In the next subsection, we analyze consumer purchase decisions and the profit-maximizing model for service providers from the perspective of the original consumer type.

## Consumers' purchase decision

To formulate the consumer purchase decision and profit maximization, we use the original consumer type. We assume that the consumers face a monopolistic service provider offering a three-part tariff scheme comprising a fixed fee (T), unit price ($p_0$), and usage allowance ($q_0$) under a fixed fee. We first define a consumer's willingness to pay (i.e., gross consumer surplus) when a consumer decide to subscribe a service and consume a specific quantity or units of service. Below, we present two distinct cases for determining willingness to pay for consuming q units of service, which may be specified by rewriting the customers' demand function as p = $[(O+r)-q]\lambda$. For consumers with preference parameter (O+r), willingness to pay can be defined as follows:

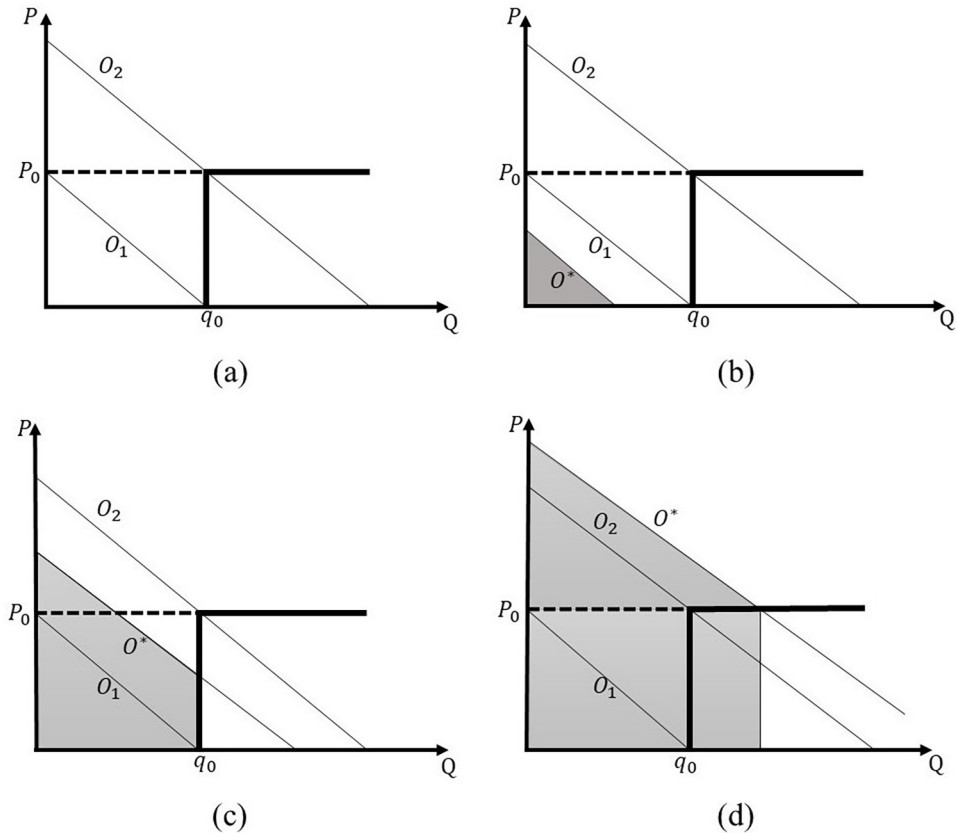

**Fig 1. Three group of consumers and its gross consumer surplus for consumers with different O*s.**

I. When $q \geq O + r$,

$$\mathrm{WTP(q)} = \int_0^{O+r} \lambda[(O + r) - t]dt$$

$$= \frac{1}{2}\lambda(O + r)^2; \tag{3}$$

II. When $q < O + r$,

$$\mathrm{WTP(q)} = \int_0^q \lambda[(O + r) - t]dt$$

$$= (O + r)\lambda q - \frac{1}{2}\lambda q^2. \tag{4}$$

To formulate the consumer's willingness to pay (i.e., gross consumer surplus), we initially assume that the probability density of the original consumer type (O) adheres to a continuous uniform distribution spanning a range of 0–1. Under specific $\lambda$, consumers with a type or preference that aligns with this uniform distribution can be categorized into three groups, based on two indices (i.e., two cut-off points): $O_1$ and $O_2$. These indices are linked to distinct three-part tariffs (i.e., associated with specific sets of $(T, p_0, q_0)$).

Particularly, the value of $O_1$ equals $q_0$, representing consumers of this type who will consume $q_0$ units when both the fixed fee (T) and unit price ($p_0$) are set at 0. The value of $O_2$ equals $q_0 + \frac{1}{\lambda} p_0$, representing consumers of this type who will consume $q_0$ units when the fixed fee (T) remains 0, but the unit price ($p_0$) exceeds 0. This implies that if a consumer's type surpasses $O_2$, they are inclined to consume additional units (beyond the allowance) by paying a unit price of $p_0$, albeit only if the fixed fee is waived. Consequently, $O_2$-type consumers exhibit greater demand compared to $O_1$-type consumers.

Under the three-part tariff pricing scheme, the monopolistic service provider deals with three endogenous variables. The amount of the fixed fee (T) will determine whether or not the consumers will subscribe to the service. If the fixed fee (T) exceed the gross surplus derived from the consumer consumption, the consumers will not choose to subscribe. In such instances, the service provider can adjust the fixed fee (T), unit price ($p_0$), and usage allowance ($q_0$) to both optimize their profits and cater to consumer demands or requirements.

Fig 1 illustrates the three consumer groups determined by the consumer indices $O_1$ and $O_2$. In Fig 1(B), the first consumer group is portrayed—these are consumers whose preferences (or types) fall within the interval $(0, O_1)$. Since $O_1$ represents consumers who would consume $q_0$ units under the condition of both zero fixed fee (T) and zero unit price ($p_0$), those with preferences lower than $O_1$ consume less than the designated usage allowance ($q_0$). This occurs even when the unit price is zero and despite their option to consume the full usage allowance. Specifically, consumers within this group, when overconfident, still opt not to fully utilize the usage allowance defined by the tariff in their actual purchase behavior.

Fig 1(C) portrays the second consumer group, whose preferences span from $(O_1, O_2)$, and who consume precisely the usage allowance upon subscribing. This group refrains from exceeding the usage allowance as the additional benefit or utility gained is less than the unit price.

Finally, Fig 1(D) illustrates the third consumer group, with preferences ranging from $(O_2, 1)$, who surpass the usage allowance. The consumption of this group deliberately exceeds the usage allowance, as the additional consumer surplus derived from consuming one unit of service surpasses the unit price. Consequently, they continue consumption until the marginal benefit is equal to the unit price.

While Fig 1 yields valuable insights into consumer preferences and decisions, it's important to recognize that the efficacy of different tariff structures can fluctuate based on market conditions and consumer preferences. Therefore, service providers should use these insights as an initial guide for devising tariff structures and determining optimal pricing strategies tailored to their particular markets. In the Section of Numerical Analysis, we offer a numerical analysis showcasing how service providers can explore optimal pricing structures under specific market conditions.

The gross consumer surplus (willingness to pay) can be used to formulate the model of whether or not the consumers will purchase a service and the amount of service units that will be consumed, depending on the three-part tariff offered. We let CS(O) be the function of the gross consumer surplus, with the overconfidence preference represented by the shaded area in Fig 1. Specifically, $CS(O_1) = \frac{1}{2} \lambda (O_1 + r)^2$ is related to Eq (3) of consumer surplus when the overconfident consumer (O+r) preference is less than the allowance $q_0$. In addition, $CS(O_2) = (O_2 + r)\lambda q - \frac{1}{2} \lambda (O_1)^2$ is related to Eq (4) of consumer surplus when the overconfident consumer (O+r) preference is larger than the allowance $q_0$. As it is known, whether or not the consumers will subscribe to the service depends on the fixed fee (T). Henceforth, O* will represent

the original consumer type, which is indifferent about subscribing or not subscribing to the service.

When T is less than $CS(O_1)$, the $O_1$ consumers will subscribe to the service, and the original consumer type $O^*$ will be less than $O_1$. We divide all the consumers subscribing to the service into three parts. First, the consumers within the interval $(O^*, O_1)$ consume O, that is, the quantity when the unit price is equal to 0. Second, the consumers within the interval $(O_1, O_2)$ consume the exact usage allowance $q_0$. Finally, the consumers within the interval $(O_2, 1)$ consume $O - \frac{1}{\lambda} p_0$. When T is between $CS(O_1)$ and $CS(O_2)$, the $O_1$ consumers will not subscribe to the service, and the original consumer type $O^*$ will be within the interval $(O_1, O_2)$. Then, we divide the consumers into two parts. First, the consumers within the interval $(O_1, O_2)$ consume the exact amount of allowance $q_0$. Second, the consumers within the interval $(O_2, 1)$ consume $O - \frac{1}{\lambda} p_0$. Theoretically, we do not need to consider the case in which T is larger than $CS(O_2)$, because the consumers will not consume any service unit if the fixed fee T is greater than the consumer surplus, in other words, the consumer will only purchase a service unit if the consumer surplus is non-negative. Therefore, we focus mainly on the scenarios where the service is feasible and consumers are willing to consume the service, i.e., the cases when T is less than $CS(O_1)$ and when T is between $CS(O_1)$ and $CS(O_2)$.

## Profit maximization model

In this section, we present the formulation of the optimization problem of the monopolistic service provider. If only a single three-part tariff pricing scheme is considered, then the profit will be composed of two elements: the fixed fee paid by the consumers subscribing to the service (T) and the price paid by the consumers for purchasing units of service exceeding the basic amount ($p_0$). We let N be the total number of consumers subscribing to the service, which can be obtained by integrating the density function f(O) from $O^*$ to 1. Thus, N is equal to $\int_{O^*}^1 f(O) dO = \int_{O^*}^1 dO = 1 - O^*$. We let Q be the expected unit consumption for all the consumers and Qe be the expected unit consumption in excess of the allowance, which are both affected by the amount of the fixed fee (T). We construct the expected profit function of the monopolist service provider as follows:

$$\Pi = T \cdot N + p_0 \cdot Qe - mc \cdot Q. \tag{5}$$

The objective function of Eq (5) is to maximize the expected total profits $\Pi$. The expected total profits is the sum of the profits from the total number of consumers (N) subscribing to the service by charged the fixed fee (T) and the usage price for purchasing units of service exceeding the basic amount $p_0$ Qe minus the marginal costs of the expected total consumption mc Q. The decision variables of this optimization problem are T, $p_0$, and $q_0$, which represent the price rate of the fixed fee, usage price, and allowance, respectively.

In the next section, we derive the profit maximization of a single three-part tariff with overconfident consumers based on two possible profit maximization cases, that is, when T is less than $CS(O_1)$ and when it is between $CS(O_1)$ and $CS(O_2)$.

**Profit maximization when the fixed fee (T) is less than $CS(O_1)$.** When the fixed fee (T) is less than $CS(O_1)$, we assume that the original consumer type $O^*$ is less than $q_0$, and the surplus of the original consumer type is related to the second case of willingness to pay in Eq (3), where $CS(O_1) = \frac{1}{2} \lambda (O^* + r)^2$. Therefore, $T = \frac{1}{2} \lambda (O^* + r)^2$. In terms of the expected unit consumption (Q), the cutoff point of the overconfidence model is based on the expected preferences or expected demand. However for the quantity consumption is based on the original

consumer type model of the three-part tariff. The service is purchased by all the consumers within the interval (O*,1); thus, the expected consumption Q is represented as follows:

$$Q = \int_{O^*}^{O_1} O*f(O)dO + q_0* \int_{O_1}^{O_2} f(O)dO + \int_{O_2}^{1} \left(O - \frac{p_0}{\lambda}\right)f(O)dO$$

$$= \frac{1}{2} + \frac{O_1^2}{2} - \frac{p_0(1 - O_2)}{\lambda} - \frac{O_2^2}{2} + q_0(-O + O_2) - \frac{(O^*)^2}{2} \tag{6}$$

Only the consumers within the range ($O_2$,1) consume more than the allowance. Thus, Qe is represented as

$$Qe = \int_{O_2}^{1} \left(O - \frac{p_0}{\lambda} - q_0\right)dO = \frac{1}{2} + \left(-\frac{p_0}{\lambda} - q_0\right)(1 - O) - \frac{O_2^2}{2} \tag{7}$$

To maximize profits, the monopolistic service provider will adjust $T$,$p_0$, and $q_0$. Profits can be classified into two types, that is, from the fixed fee and from the usage price, as shown in Eq (8).

$$\begin{aligned} \text{Profit} &= T*N + p_0*Q_e - mc*Q \\ &= p_0 \left(\frac{1}{2} + \left(-\frac{p_0}{\lambda} - q_0\right)(1 - O_2) - \frac{O_2^2}{2}\right) + T(1 - O^*) \\ &\quad - mc \left(\frac{1}{2} + \frac{O_1^2}{2} - \frac{p_0(1 - O_2)}{\lambda} - \frac{O_2^2}{2} + q_0(-O_1 + O_2) - \frac{(O^*)^2}{2}\right) \end{aligned} \tag{8}$$

Solution detail of Eqs (6), (7) and (8) are available in S1 Appendix. Furthermore, we differentiate the profit function with respect to $T$,$p_0$, and $q_0$ to determine the optimal value of each endogenous variable. The Equations of first-derivative conditions and the substitution Equations to obtain the best solution are represent in S1 Appendix. We obtain the maximum expected profit by including the optimal values of the endogenous variables into Eq (8). Thus,

we obtain the expected profit of the service provider, as follow:

$$\text{Profit} = \frac{1}{2\lambda^2}\left(\frac{1}{27}(-mc + 2\sqrt{mc^2 + r^2\lambda^2} - \sqrt{5mc^2 + 4r^2\lambda^2 - 4mc\sqrt{mc^2 + r^2\lambda^2}})^3\right.$$

$$-\frac{1}{9}(-mc + 2\sqrt{mc^2 + r^2\lambda^2}$$

$$-\sqrt{5mc^2 + 4r^2\lambda^2 - 4mc\sqrt{mc^2 + r^2\lambda^2}}^2\left(mc + 2\lambda - 2\lambda\left(1 + \frac{mc}{\lambda} - \frac{\sqrt{mc^2 + r^2\lambda^2}}{\lambda}\right)\right)$$

$$+\frac{1}{3}\lambda\left(-mc + 2\sqrt{mc^2 + r^2\lambda^2} - \sqrt{5mc^2 + 4r^2\lambda^2 - 4mc\sqrt{mc^2 + r^2\lambda^2}}\right)$$

$$\left(2mc + \lambda - r^2\lambda - 2(mc + \lambda)\left(1 + \frac{mc}{\lambda} - \frac{\sqrt{mc^2 + r^2\lambda^2}}{\lambda}\right) + \lambda\left(1 + \frac{mc}{\lambda} - \frac{\sqrt{mc^2 + r^2\lambda^2}}{\lambda}\right)^2\right)$$

$$+\lambda\left(\frac{1}{9}(1 + r)\lambda\left(4mc - 2mcr + \frac{2mc^2}{\lambda} + 2\lambda + 4r\lambda + 2r^2\lambda + \sqrt{(-36mc^2r^2}\right.\right.$$

$$+\left(-4mc + 2mcr - \frac{2mc^2}{\lambda} - 2\lambda - 4r\lambda - 2r^2\lambda\right)^2)) - \frac{1}{27}\sqrt{\lambda}\left(4mc - 2mcr + \frac{2mc^2}{\lambda} + 2\lambda\right.$$

$$+4r\lambda + 2r^2\lambda + \sqrt{\left(-36mc^2r^2 + \left(-4mc + 2mcr - \frac{2mc^2}{\lambda} - 2\lambda - 4r\lambda - 2r^2\lambda\right)^2\right)}^{3/2}$$

$$+mc\left((-1 + 2r^2)\lambda - \frac{2}{3}r\sqrt{\lambda}\sqrt{\left(4mc - 2mcr + \frac{2mc^2}{\lambda} + 2\lambda + 4r\lambda + 2r^2\lambda\right.}\right.$$

$$+\sqrt{\left(-36mc^2r^2 + \left(-4mc + 2mcr - \frac{2mc^2}{\lambda} - 2\lambda - 4r\lambda - 2r^2\lambda\right)^2\right)}\right)$$

$$+\frac{1}{9}\left(4mc - 2mcr + \frac{2mc^2}{\lambda} + 2\lambda + 4r\lambda + 2r^2\lambda + \sqrt{(-36mc^2r^2}\right.$$

$$+\left(-4mc + 2mcr - \frac{2mc^2}{\lambda} - 2\lambda - 4r\lambda - 2r^2\lambda\right)^2\right)\right)\right)\right) \quad (9)$$

**Profit maximization when $CS(O_1) < T < CS(O_2)$.** When the fixed fee (T) is within the interval$(S(O_1), S(O_2))$, the marginal preference of the original consumer $O^*$ is within the range $(O_1, O_2)$, and the surplus of the marginal consumer is related to the first case of willingness to pay in Eq (4), where $CS(O_2) = (O^* + r)\lambda q_0 - \frac{1}{2}\lambda(q_0)^2$. Thus,
$T = (O^* + r)\lambda q_0 - \frac{1}{2}\lambda(q_0)^2$. The service is purchased by all the consumers within the interval

(O*,1); thus, the expected consumption Q is represented as

$$Q = q_0 * \int_{O^*}^{O_2} f(O)dO + \int_{O_2}^{1}\left(O - \frac{p_0}{\lambda}\right)f(O)dO = \frac{1}{2} - \frac{p_0(1 - \theta_2)}{\lambda} - \frac{\theta_2^2}{2} + q_0(\theta_2 - \theta^*) \quad (10)$$

Only the consumers within the range $(O_2,1)$ consume more than the allowance; thus, Qe is represented as

$$Qe = \int_{O_2}^{1}\left(O - \frac{p_0}{\lambda} - q_0\right)dO = \frac{1}{2} + \left(-\frac{p_0}{\lambda} - q_0\right)(1 - \theta_2) - \frac{\theta_2^2}{2} \quad (11)$$

To maximize the expected profit, the monopolistic service provider can adjust $T, p_0$, and $q_0$. As stated above, the two types of profits are from the fixed fee and from the usage price, as shown in Equation (12).

$$\text{Profit} = T*N + p_0*Q_e - mc*Q$$

$$= p_0\left(\frac{1}{2} + \left(-\frac{p_0}{\lambda} - q_0\right)(1 - \theta_2) - \frac{\theta_2^2}{2}\right) - mc\left(\frac{1}{2} - \frac{p_0(1 - \theta_2)}{\lambda} - \frac{\theta_2^2}{2} + q_0(\theta_2 - \theta^*)\right)$$
$$+ T(1 - \theta^*) \quad (12)$$

The solution details of Eqs (10), (11), and (12) are available in S1 Appendix. To obtain the profit-maximizing values of the endogenous variables, we differentiate the profit function with respect to $T, p_0$, and $q_0$. The equations of first-derivative conditions are represent in S1 Appendix. Based on the first-derivative condition, the result shows that the profit function takes the extreme value of the first-order differential of $q_0$, which is limited in obtaining the optimal solution. The result of the first-order differential of $q_0$ is demonstrated As follows:

$$\frac{\partial \Pi}{\partial q_0} = -p_0(1 - \theta_2) - mc(\theta_2 - \theta^*) = -mcr + \frac{T}{2} + \frac{p_0^2}{\lambda} + \frac{T^2}{\lambda q_0^2} - 2mcq_0 + p_0\left(-\frac{mc + \lambda}{\lambda} + q_0\right)$$
$$< 0 \quad (13)$$

As the Eq (13) shows a negative result, it implies that profits can be increased by reducing the usage provided under fixed fees. However, if the allowance is reduced too much, consumers may not be willing to consume at least by the amount of $q_0$. This goes against the assumption that the fixed fee (T) falls within the interval $(S(O_1), S(O_2))$, and such a scenario cannot exist in this model.

## Mathematical model of the three-part tariff pricing with consumer overconfidence

In this part, we examine the design of three-part tariff pricing plans after formulating the analytic model in the previous section. A mathematical model is formulated to determine the profit-maximizing tariffs. In particular, numerical analysis using a genetic algorithm is carried out to derive the optimal single three-part tariff and to compare other menus of three-part tariffs (two-and three-optional) to analyze the performance of different tariff structures and derive the guidelines for determining the best timing of using the different tariff structures. We utilize index i to represent the ith consumer and indexes j and j′ to represent the jth and j′th tariff among the available set J of tariff structures that contain multiple three-part tariffs that can be chosen by the consumer. The mathematical model of the three-part tariff pricing across the total number of consumers (I) for the optimization problem of the monopolistic

service provider can be constructed as follows:

$$\max \pi(T_J, p_{0J}, q_{0J}) = \sum_{i \in I} \sum_{j \in J} [T_j + p_{0j}.qe_{i,j}(T_j, p_{0j}, q_{0j}) - q_{i,j}(T_j, p_{0j}, q_{0j}).c].w_{i,j} \tag{14}$$

subject to

$$CS_{i,j}(T_j, p_{0j}, q_{0j}).w_{i,j} \geq 0 \qquad (i \in I, j \in J), \tag{15}$$

$$(CS_{i,j}(T_j, p_{0j}, q_{0j}) - CS_{i,j'}(T_{j'}, p_{0j'}, q_{0j'})).w_{i,j} \geq 0 (i \in I, j, j' \in J \wedge j > j'), \tag{16}$$

$$(CS_{i,j}(T_j, p_{0j}, q_{0j}) - CS_{i,j'}(T_{j'}, p_{0j'}, q_{0j'})).w_{i,j} < 0 (i \in I, j, j' \in J \wedge j < j'), \tag{17}$$

$$T_j, p_{0j}, q_{0j} \geq 0, \quad (j \in J), \text{ and} \tag{18}$$

$$w_{i,j} = \{0, 1\} \ (i \in I, j \in J). \tag{19}$$

The objective function is Eq (14), in which the service provider seeks to maximize its total expected profits ($\pi$). As the sum of the profits resulting from all (I) consumers, total expected profits are composed of three elements: fixed fee ($T_J$) paid by the consumers who subscribe to the service, price consumers paid for purchasing extra units of service that exceed the basic amount $p_{0j}.qe_{i,j}(T_j, p_{0j}, q_{0j})$, and the cost of the total expected consumption $q_{i,j}(T_j, p_{0j}, q_{0j}) \cdot c$. Constraint (15) is the individual rationality or participation constraint, which can only be satisfied when the net consumer surplus $CS_{i,j}(T_j, p_{0j}, q_{0j})$ multiplied by the binary decision variables $w_{i,j}$ is larger or equal to zero. This participation constraint ensures that a consumer with a negative net consumer surplus does not subscribe to the service. Other than individual rationality constraints that need to be satisfied, given the availability of a tariff menu with numerous choices, the incentive compatibility constraints in Equations (16) and (17) also need to be satisfied. The equations distinguish whether the menu of jth tariff is larger than that of j'th tariff, to ensure that the consumer selects the tariff menu with the greatest consumer surplus. if two or more tariff menus produce the same consumer surplus, then they choose no more than one tariff. $T_j, p_{0j}$, and $q_{0j}$ are the decision variables of this optimization problem, and are continuous, non-negative variables that represent the price rate as mentioned in constraint (18). In constraint (19), the binary variable $w_i$ equals 1 if the ith consumer chooses to subscribe to the service, and 0 otherwise. Next, we discuss each element of the total profit function and constraint as follows: usage quantity $q_{i,j}(T_j, p_{0j}, q_{0j})$, usage exceeding the allowance $qe_{i,j}(T_j, p_{0j}, q_{0j})$, and consumer surplus $CS_{i,j}(T_j, p_{0j}, q_{0j})$.

For different consumers, the decision about the usage quantity $q_i(T, p_0, q_0)$ depends mainly on the consumer type (i.e., preference O) and the price rate of the three-part tariff. Consumers can be divided into three types based on the magnitude of O. According to the arguments in the section related to consumer's purchase decision, the usage quantity of the individual

consumers can be represented by:

$$
q_{i,j}\left(T_j, p_{0j}, q_{0j}\right), = \begin{cases}
O & \text{if } O < O_1, T < \frac{1}{2}\lambda(O+r)^2 \\[2mm]
0 & \text{if } O < O_1, T > \frac{1}{2}\lambda(O+r)^2 \\[2mm]
q_0 & \text{if } O_1 \le O \le O_2, T < \lambda(O+r)q_0 - \frac{1}{2}\lambda q_0^2 \\[2mm]
0 & \text{if } O_1 \le O \le O_2, T > \lambda(O+r)q_0 - \frac{1}{2}\lambda q_0^2 \\[2mm]
O - \frac{1}{\lambda}p_0 & \text{if } O > O_2, T < \frac{1}{2}\left(\lambda(O+r)+p_0\right)\left(O+r-\frac{1}{\lambda}p_0\right) \\[2mm]
0 & \text{if } O > O_2, T > \frac{1}{2}\left(\lambda(O+r)+p_0\right)\left(O+r-\frac{1}{\lambda}p_0\right)
\end{cases}
\tag{20}
$$

Based on the expression of the usage quantity, $qe_i(T, p_0, q_0)$, extra usage units exceeding the allowance for overconfidence in consumer are expressed as follows:

$$
qe_{i,j}\left(T_j, p_{0j}, q_{0j}\right) = \begin{cases}
O - \frac{1}{\lambda}p_0 - q_0 & \text{if } O > O_2 \\[2mm]
0 & \text{if } O \le O_2
\end{cases}
\tag{21}
$$

The consumer surplus $CS_i(T, p_0, q_0)$ is defined as the difference between the willingness-to-pay and the total price paid by the overconfidence consumer in the three-part tariff. Eq (22) shows the consumer surplus $CS_{i,j}(T_j, p_{0j}, q_{0j})$ for the three possible cases:

$$
CS_i\left(T, p_0, q_0\right) = \begin{cases}
\frac{1}{2}\lambda(O+r)^2 - T & \text{if } O < O_1, T < \frac{1}{2}\lambda(O+r)^2 \\[2mm]
0 & \text{if } O < O_1, T > \frac{1}{2}\lambda(O+r)^2 \\[2mm]
(O+r)\lambda q_0 - \frac{1}{2}\lambda q_0^2 - T & \text{if } O_1 \le O \le O_2, T < \lambda(O+r)q_0 - \frac{1}{2}\lambda q_0^2 \\[2mm]
0 & \text{if } O_1 \le O \le O_2, T > \lambda(O+r)q_0 - \frac{1}{2}\lambda q_0^2 \\[2mm]
\frac{1}{2}\left(\lambda(O+r)+p_0\right)\left(O+r-\frac{1}{\lambda}p_0\right) - T - \left(O+r-\frac{1}{\lambda}p_0 - q_0\right)p_0 & \text{if } O > O_2, T < \frac{1}{2}\left(\lambda(O+r)+p_0\right)\left(O+r-\frac{1}{\lambda}p_0\right) \\[2mm]
0 & \text{if } O > O_2, T > \frac{1}{2}\left(\lambda(O+r)+p_0\right)\left(O+r-\frac{1}{\lambda}p_0\right)
\end{cases}
\tag{22}
$$

## Comparative statics analysis: Effect of variable cost, price sensitivity, and overconfidence level on the optimal solution

To understand the effect of the pricing combination on the optimal solution when each parameter (exogenous variable) in the model changes, we conduct a comparative statics analysis similar to a numerical analysis of the optimal solutions from the tariff combinations obtained in the previous section. In this section, we mainly analyze the changes in the endogenous variables (charge per unit [$p_0$], fixed fee [T], unit allowance provided under the fixed fee [$q_0$], and expected profit [$\pi$]) with respect to the exogenous variables: overconfidence level (OL), variable costs (mc), and consumer price insensitivity ($\lambda$), through numerical examples

**Table 1. Comparative statics analysis on parameters.**

| Parameter | | | Optimal values | | | |
|---|---|---|---|---|---|---|
| Lambda | mc | Overconfidence level (OL) | | T | $p_0$ | $q_0$ | Profit |
| Small: 0.5 | Low: 0.05 | Very low | 0.1 | 0.156 | 0.06 | 0.958 | 3.55 |
| | | Low | 0.2 | 0.181 | 0.115 | 0.874 | 5.97 |
| | | Medium | 0.3 | 0.207 | 0.177 | 0.783 | 6.72 |
| | | High | 0.4 | 0.236 | 0.241 | 0.687 | 8.82 |
| | | Very High | 0.5 | 0.267 | 0.306 | 0.590 | 11.29 |
| | High: 0.1 | Very low | 0.1 | 0.181 | 0.082 | 0.976 | 2.37 |
| | | Low | 0.2 | 0.204 | 0.121 | 0.917 | 3.72 |
| | | Medium | 0.3 | 0.229 | 0.173 | 0.839 | 5.46 |
| | | High | 0.4 | 0.257 | 0.231 | 0.752 | 7.60 |
| | | Very High | 0.5 | 0.286 | 0.292 | 0.661 | 10.18 |
| Large: 1 | Low: 0.05 | Very low | 0.1 | 0.290 | 0.115 | 0.938 | 8.43 |
| | | Low | 0.2 | 0.340 | 0.241 | 0.843 | 11.33 |
| | | Medium | 0.3 | 0.394 | 0.372 | 0.745 | 14.83 |
| | | High | 0.4 | 0.388 | 0.504 | 0.646 | 18.96 |
| | | Very High | 0.5 | 0.517 | 0.636 | 0.547 | 23.77 |
| | High: 0.1 | Very low | 0.1 | 0.313 | 0.121 | 0.958 | 7.10 |
| | | Low | 0.2 | 0.362 | 0.231 | 0.876 | 9.95 |
| | | Medium | 0.3 | 0.415 | 0.354 | 0.783 | 13.45 |
| | | High | 0.4 | 0.472 | 0.483 | 0.687 | 17.65 |
| | | Very High | 0.5 | 0.535 | 0.613 | 0.590 | 22.58 |

that may affect the optimal solution. Table 1 presents the result of comparative statics analysis for different parameters.

Subsequently, we review and evaluate the effects of the changes in the various parameters on the optimal values of the decision variables. Fig 2 shows a diagram of the results of the comparative statics analysis of the optimal solution when the exogenous parameters change. The horizontal axis indicates different overconfidence levels; the vertical axis displays the optimal fixed fee, usage price, and usage allowance; and the secondary vertical axis represents the optimal profit.

## Impact of changes in λ on optimal solution

Fig 2 shows the comparative static analysis of the optimal three-part pricing when the consumer's preference pattern has a uniform distribution. First, we analyze the effects of change in parameter $\lambda$ on the optimal solution. We examine several values of $\lambda$ as follows: $\lambda = 0.5$ for low and $\lambda = 1$ for high. Fig 2.1–2.4 show that as $\lambda$ increases, all endogenous variables (unit price ($p_0$), fixed fee ($T$), unit provided under the fixed fee ($q_0$) and profits ($\pi$) also increase. The reason is that a large parameter $\lambda$ leads to less price sensitivity. These types of consumers are more willing to pay more for the product or service, and thus drive the service provider to increase the charge per unit ($p_0$), fixed fee ($T$), and the unit provided under the fixed fee ($q_0$) to preserve sales volume. In general, as shown in Fig 2.1 and 2.2, when consumers have high price sensitivity (or small $\lambda$), the service providers face the risks of losing customers, which can result in reduced the expected profits. After all, a smaller price sensitivity can increase the price component and, naturally, the profit.

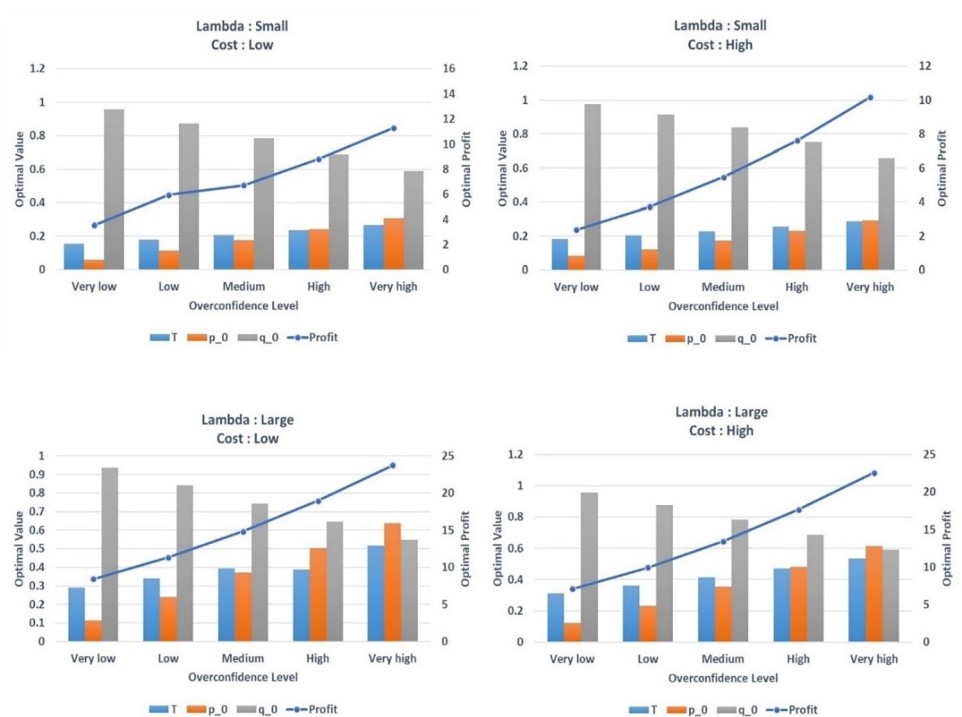

**Fig 2. Results of comparative statics analysis of the optimal solution when the exogenous parameter changes.**

### Impact of changes in mc on optimal solution

Second, we investigate the impact of changes at several values of *mc* as follows: *mc* = 0.05 for low level and *mc* = 0.1 for high level on the optimal solutions. Fig 2.1–2.4 show that increasing the variable cost *mc* also increases fixed fee ($T$), the unit provided under the fixed fee ($q_0$) and the unit price ($p_0$). However, an increase in variable cost *mc* reduces profits ($\pi$). In other words, when the variable cost *mc* is high, the fixed fee ($T$) and usage provided are also high. Thus, increasing the variable cost also increases the price to cover such cost and increase profit.

### Impact of changes in overconfidence level on optimal solution

We also examine diverse estimations of the overconfidence level (OL) for the optimal solutions, and the reasonable level of overconfidence is between 0.1 and 0.5. Specifically, we set five levels of overconfidence, that is, OL = 0.1 for very low, OL = 0.2 for low, OL = 0.3 for medium, OL = 0.4 for high, and OL = 0.5 for very high. When considering the reasonable range of the overconfidence level, several previous studies considered using the similar values (see, for example, Grubb [21] and Campbell et al. [29]. Fig 2.1–2.4 show that an increase in the overconfidence level (OL) will cause an increase in the fixed fee (T), unit price ($p_0$), expected profits ($\pi$) and will cause decrease in unit allowance provided under the fixed fee ($q_0$). When the overconfidence level (OL) is high, the fixed fee (T) and the charge per unit ($p_0$) will increase to mitigate costs and increase profits, and unit allowance provided under the fixed fee ($q_0$) will decrease, accordingly.

### Numerical analysis

In this section, given the complexity of formulas, we carry out a numerical analysis to derive the optimal single three-part tariff and to compare other menus (two- and three-optional) to

analyze the performance of different tariff structures and drive the guidelines for determining the best timing of the use of different tariff structures. Furthermore, due to increasing the number of users and pricing structures, the complexity requires more binary variables and thus generates a large-scale combinatorial problem, which has NP-complete computational complexity [30]. Thus, we decide to resort to genetic algorithm to tackle the computational issue. Even though, the use of a genetic algorithm may not guarantee the global optimal solution, but we follow a series of steps to carefully check the efficiency of the algorithm, which should be able to achieve nearly global optimal results to enable us to derive some useful implications.

## Genetic algorithm

The natural selection process described by Holland [31] in biology served as the basis for the development of the genetic algorithm (GA), which is a metaheuristic technique renowned for its ability to handle large search areas. Implementing a real-valued GA in many systems would be straightforward [32, 33]. A population is first generated randomly by this algorithm. A group of individual solutions can be employed to characterize the population (chromosomes). In our model, these chromosomes comprise the fixed fee, the usage price and the allowance (i.e., the three key components of the three-part tariff). They are randomly generated from a uniform distribution to ensure the initial population of individuals possesses diversity, with a broad range of values for each component. The algorithm aims to find the optimal combination of these tariff components that maximizes profit by iteratively evolving the chromosomes across multiple generations. This progression involves the process of selection, crossover, and mutation, wherein chromosomes evolve and give rise to new offspring. This mechanism emulates the principles of natural selection, where the most adaptable individuals are likelier to pass on their genetic material to the next generation.

Within the selection process, a small subset of solutions from the current population is chosen by the algorithm to act as parents, based on the parents' fitness value, to form the basis for a new generation based on the parents' fitness value. A chromosome is randomly selected from the population, its probability determined by the fitness function. Genetic mechanisms such as crossover and mutation generate the new generation. Once the termination condition is met, the best chromosome from the latest generation is presented as the final solution [34].

The number of generations, population size, crossover rate, and mutation rate are the important parameters typically included in GA programs to define and control the four core genetic operators associated with the algorithm [35]. The GA produces an efficient, optimal, or a nearly optimal solution depending on the parameters. The four important GA parameters we employ in our study are summarized below.

1. Generation: This parameter determines the total number of cycles (generations) necessary for the algorithm to complete. This study employs a termination criterion based on a maximum of 1,000 generations. During each generation, the algorithm assesses the fitness of each chromosome (tariff structure) based on its profit generation. Chromosomes with higher fitness hold a greater likelihood of selection for reproduction. Here, the individual with the best fitness value across 1,000 generations is identified as the optimal solution.

2. Population size: 50 chromosomes are randomly selected as the population of this study. A population's size reveals details about its structure. According to De Jong and Spears [36], the ideal population size should fall between 50 and 100.

3. Crossover rate: The crossover rate denotes the likelihood of applying crossover (or recombination) to a pair of selected parent chromosomes to generate offspring chromosomes.

Within the chromosome, where parental genetic material exchange occurs, a random gene is selected for crossover implementation. Subsequently, a new offspring is produced via crossover, contingent upon the chosen crossover point and specific parent components. We employ a real-value crossover in this study to choose the point at random, which then transfers the necessary components from the two parents to produce an offspring. The offspring can be determined as follows:

$$o_i = \alpha \cdot x_i + (1 - \alpha) \cdot y_i, \tag{23}$$

where $\alpha$ is a random number with a uniform distribution between 0 and 1, and $x_i$ and $y_i$ are the two chosen gene-pool chromosomes.

4. Mutation rate: The mutation rate refers to the probability that a gene within a chromosome will undergo a mutation during the evolution process. Mutation often occurs after the completion of the crossover. The operator generates new adaptive solutions and helps prevent local optima by applying the modifications arbitrarily to one or more "genes" to create new offspring. Similar to a crossover, real-value mutation [37] is a method for producing new offspring from parents. The following can be used to determine the mutation rate:

$$z_i = x_i + \text{sign} \cdot r_i \cdot \delta. \tag{24}$$

Sign$\in\{+1,-1\}$ is chosen with equal probability, $r_i = \rho(\text{range}_i)$, $\rho \in [0.1, 0.5]$. The range ratio ($\rho$), which is also known as the variable range ratio, is related to the maximum step that a mutation is permitted to produce; $\text{range}_i$ is the range of variable $x_i$; and the parameter $\delta$ indicates the extent to which offspring can be produced from the parents and is expressed as follows:

$$\delta = 2^{-k.u}, \tag{25}$$

where $k \in N^+$, and $u \in [0,1]$, with uniform probability. The mutation precision level necessary to reach the optimum depends on parameter $k$. However, in practice, the value of $k$ is determined by the expected value of the mutation steps, and the higher the value of $k$, the more precise the resulting mutation operator. The GA operators employed in this investigation are described in detail below. To update the solution, we set $\rho$ to 0.1, signifying a low mean mutation step. Moreover, we set $k$ to 10, which complements the approach and is known to have a significantly strong influence on performance.

## Comparison of the analytical model and the genetic algorithm on the optimal solution

In this section, we carry out a numerical example when the genetic algorithm is used to drive the optimal solution for different overconfidence levels. The results are compared with the optimal case of the analytical model. We build the mathematical model without any restriction to find the optimal solution, but for the analytical model, we use case 1 as found to be the optimal solution. In this example, the sample comprises 100 consumers that follow a uniform distribution and assume that the parameters are as follows: overconfidence level = 0.1, 0.2, 0.3, 0.4, 0.5; $\lambda = 1$; mc = 0.1. Table 2 shows the computational results of the optimal solution

Table 2 shows that the genetic algorithm provides an excellent approximation of the optimal solution of the objective value (the expected profit). The resulting profits is very similar to the analytical results. The genetic algorithm can achieve an excellent approximation of the optimal solution and can search the solution space efficiently even for this 100 samples of

**Table 2. Comparison of the optimal solutions between analytical model and genetic algorithm.**

| Case | Overconfidence level | Profit |
|---|---|---|
| Analytical Model | 0.1 | 7.10 |
| | 0.2 | 9.95 |
| | 0.3 | 13.45 |
| | 0.4 | 17.65 |
| | 0.5 | 22.58 |
| Genetic Algorithm | 0.1 | 7.10 |
| | 0.2 | 9.96 |
| | 0.3 | 13.45 |
| | 0.4 | 17.66 |
| | 0.5 | 22.57 |

consumers, and is thus used in the analysis to compare the performance of different menu of three-part tariff pricing structures.

## Comparison of different pricing structures

Better pricing strategies have been demonstrated to effectively segment the market. Offering a broader range of tariffs to consumers may also aid the service provider in catering to diverse needs and preferences, ultimately leading to higher customer satisfaction and loyalty, and potentially augmented revenue and profits. However, it's important to acknowledge that providing a greater number of tariffs might also incur higher administrative or marketing costs.

Within this section, we undertake an analysis and evaluation of various tariff structures. Alongside examining the profitability of different tariff designs in the presence of overconfident consumers, our analysis also seeks to investigate the influence of implementing diverse pricing structures on total user participation and overall usage quantities.

In particular, we scrutinize the performance of the two-part tariff and the menu of three-part tariffs (including single, double, and triple optional three-part tariffs) within simulated experimental scenarios. Furthermore, several parameters, related to consumer behavior and market structure, are manipulated in the experiment. Factors that could potentially impact the efficacy of tariff structures, such as consumer preferences, price sensitivity, and the degree of overconfidence, are duly considered in this study.

To encapsulate consumer diversity, a sample of 1000 consumers exhibiting varying values of $\theta$ and $\lambda$ has been employed. The parameter $\theta$, symbolizing consumer preference, is assumed to follow a triangular distribution—a flexible distribution that is relatively easy to specify, with its support ranging between zero and one. It allows for both high and low means and variances in our experiment. Additionally, to explore the influence of price sensitivity heterogeneity on tariff structure profitability, the simulation study adopts a distribution of $\lambda$ characterized by a triangular distribution with small and large variances. Although this setting might lead to demand functions for distinct consumers violating the Spence-Mirrlees single-crossing property, it better mirrors real-world situations where consumer behavior often displays significant heterogeneity. Lastly, triangular distributions are also employed to model the extent of overconfidence (with low and high means, and low and high variances). This strategy is embraced to more accurately capture the diversity in consumers' levels of overconfidence and provide a more nuanced representation of real-world market phenomena.

Finally, to bolster the robustness of analysis, each of the 25 experimental settings (i.e., combinations of different parameters) is subjected to five replications in the simulation. This culminates in 160 observations for each of the four tariff structures. The details of the

**Table 3. Experimental parameter setting.**

| Factors | Symbol | Number | Factor Levels |
|---|---|---|---|
| Mean of preference parameter | Mean | 2 | Low—High |
| Preference parameter variation | Variance | 2 | Large—Small |
| Mean of overconfidence parameter | OC Mean | 2 | Low—High |
| Variance of overconfidence parameter | OC Variance | 2 | Large—Small |
| Consumer price sensitivity | Lambda Var | 2 | Large—Small |
| Total number of environmental settings | $2^5 = 32$ | | |
| Number of replications | 5 | | |
| Total number of observations for each tariff structure | 160 | | |
| Total number of tariff structures | 4 | | |
| Total number of observations | 640 | | |

experimental design are delineated in Table 3, while Tables 4–6 furnish the distributions of consumer preferences, overconfidence levels, and price sensitivity (i.e., variability of lambda) employed in the simulation.

## Results of the experiment

To obtain the optimal pricing strategy across all tariff structures, we establish and conduct corresponding experiments entailing the simulation of various consumer distributions. The optimal pricing schemes, yielding the highest total expected profit, are determined by solving the mathematical programming problem using the genetic algorithm for each experimental setup and pricing structure. Additionally, the number of service subscribers and the total service consumption quantity linked to the pricing scheme leading to the highest expected profit are documented for subsequent analysis.

**Analysis of profitability.** To decompose the effects attributed to each individual factor on various performance measures (i.e., profitability, # of participants, and usage quantity), we employ analysis of variance (ANOVA) models. The ANOVA model outcomes are presented in Table 7. When using maximum profit obtained under different combination of experiment settings as the response variable, the results reveal significant impacts of pricing structures and experimental factors (i.e., mean and variance of overconfidence level, mean and variance of consumers' preferences, and price sensitivity) on the profit of the service provider.

Specifically, the outcomes in Table 7 indicate that profit is likely influenced concurrently by mean preference, variance of preference, and level of overconfidence. Notably, the major driver influencing profits is the mean of consumers' preferences (i.e., whether or not consumers possess a high mean preference for a service), which accounts for approximately 74% of variance in profit. This finding is consistent with previous studies that did not incorporate the overconfidence parameter (see, for example, Lin et al. [38]). Ultimately, market demand serves as a crucial profit determinant.

**Table 4. Distribution of consumer's preferences.**

| Mean | Variance | Preferences Distribution |
|---|---|---|
| High | Large | $\Delta$ (0.4, 0.7, 1) |
| High | Small | $\Delta$ (0.55, 0.7, 0.85) |
| Low | Large | $\Delta$ (0.2, 0.5, 0.8) |
| Low | Small | $\Delta$ (0.35, 0.5, 0.65) |

**Table 5. Distribution of consumer's overconfidence level.**

| Mean | Variance | OC Distribution |
|---|---|---|
| High | Large | Δ (0.1, 0.3, 0.5) |
| High | Small | Δ (0.25, 0.3, 0.35) |
| Low | Large | Δ (0.1, 0.2, 0.3) |
| Low | Small | Δ (0.15, 0.2, 0.25) |

**Table 6. Distribution of consumer's price sensitivity.**

| Lambda variance | Lambda |
|---|---|
| | Distribution |
| Small | Δ (0.75, 1.0, 1.25) |
| Large | Δ (0.5, 1.0, 1.5) |

In the present study, introducing consumers' overconfidence as an experimental factor also underscores the importance of consumers' mean overconfidence level in influencing profitability, explaining around 15% of variability in profit. Given that this finding on the primary effect of consumers' overconfidence aligns with our expectations, a service provider can enhance profits (or producer surplus) by targeting customers with high overconfidence levels.

Fig 3 illustrates that as consumers' overconfidence levels rise, profits also increase, and vice versa. In our simulation experiment, service provider's profitability experiences an approximate 40% boost when consumers' overconfidence levels are high. This finding aligns with prior research such as Grubb [21] and Li et al. [12], which demonstrate an improvement in the service provider's profit when consumer overconfidence is high.

While it's intuitive that the degree of overconfidence in consumer preferences or behavior directly impacts service provider profits, the influence of overconfidence in tariff probability or performance is moderated by other conditions or factors. The true intrigue lies in how these boundary conditions affect tariff performance and yield valuable managerial implications.

It is crucial to emphasize that assessing solely the main effects of consumers' preferences and overconfidence is insufficient when interaction effects are also at play, especially in cases where interactions occur between tariff structures and market environmental factors. Specifically, the mean preferences of consumers and their overconfidence levels play significant roles in the service providers' profits for various pricing structures, as depicted in Fig 4.

In particular, when the mean preference is low and the overconfidence level is also low, a simpler tariff structure involving two price components should suffice. Consumers' low preference and a lack of significant overestimation of their demand suggest relatively homogeneous demand among most consumers. This implies that there is no need for overly complex tariff structures such as a menu of three-part tariffs. This finding aligns with Bhargava and Gangwar [39], who demonstrated that a three-part tariff can be simplified into a two-part tariff with zero allowance under certain market demand conditions.

However, it is worth noting that certain studies, such as Jang and Kwon [28] and Grubb [1], have highlighted the advantages of three-part tariffs in cases where consumers exhibit overconfidence or behavioral biases. In our study, we discovered that the menu of three-part tariffs can yield greater profitability than other pricing structures in specific combinations. Particularly, when consumers have low preferences but high overconfidence, pricing structures other than the two-part tariff appear to be superior choices. In this scenario, adopting an

alternative tariff involving a menu of three-part tariffs could increase profits by approximately 15%, as depicted in Fig 4(A).

Nevertheless, if consumers' overconfidence levels are high, then regardless of whether their preferences are high or low, adopting a two-part tariff structure is no longer a reasonable option, and the profit disparities between the two-part tariff and other pricing structures become significantly distinct, as shown in Fig 4(B).

We can delve further into examining combinations of factors to extract useful managerial implications. For instance, as depicted in Fig 5, when the mean preference is high and the level of overconfidence is also high, although a more intricate tariff structure might appear suitable in theory, other considerations still warrant attention.

Indeed, in this scenario, if consumer preferences exhibit minimal variability (i.e., low variance of preference), as shown on the right-hand side of Fig 5(A), a two-part tariff structure could still prove effective since the overestimated demand might roughly satisfy the non-decreasing price elasticity (NPE) condition, a condition discussed in Bhargava and Gangwar [39].

However, if the variance of preference is substantially high, a more intricate tariff structure would be necessary to accommodate diverse consumer needs. Specifically, when both the mean and variance of preference are high, the nonparallel lines in Fig 5(B) illustrate that with high overconfidence levels, introducing multiple tariff structures can achieve improved price discrimination and significantly influence a service provider's profit potential. The larger gap between the two lines in the figure for a higher variance of preference emphasizes this effect. In such cases, two-part tariffs result in lower profits. Tariff structures involving three-part tariffs (whether one, two, or three optional three-part tariffs) tend to yield relatively higher profits. The service provider's profitability decreases by around 13% when using a tariff with just two price components. This phenomenon becomes more pronounced when the variance of lambda (representing heterogeneity in price sensitivity) is also high.

**Analysis of total number of participants.** While the service provider's primary objective is to maximize revenues, other crucial metrics such as the number of customers who subscribe to the service and the amount of service consumed also require consideration. The second column of Table 7 presents the results of the analysis of variance when considering parameters related to consumer demand, the level of overconfidence, and tariff structure as input factors, with the total number of participants as the response variable. It becomes evident that all the (main) factors significantly affect the number of consumers subscribing to or using the service. Specifically, the results in Table 7 indicate that the variance of price sensitivity (i.e., $\lambda$) and the variance of consumer preference (i.e., O) are likely to play important roles in influencing the number of user participation, but their effects differ across various tariff structures.

Theoretically, when consumers are very homogeneous, all tariff structures might achieve a similar level of performance in attracting user participation, as shown in Fig 6. This means that when the variance of price sensitivity is small and the variance of preference is small, all tariff structures could potentially result in a higher number of users subscribing. However, when the variance of price sensitivity is large (regardless of whether variance of preference is large or small), other tariff structures will outperform the two-part tariff. The similar situation might arise when the variance of price sensitivity is small but the variance of preference is large.

Indeed, the price sensitivity of consumers, represented by $\lambda$, significantly affects the total number of participating users, particularly when interacting with the mean and variance of preference, as illustrated in Fig 7. Specifically, when the variation of $\lambda$ is small (indicating limited diversity in price sensitivity among consumers), and the mean preference is low, coupled with a high variance of preference, a two-part tariff pricing scheme fails to attract a substantial

**Table 7. Results of analysis of variance.**

| Main and interaction factors | Variance explained (%) | | |
|---|---|---|---|
| | **Profit** | **# of participants** | **Usage quantity** |
| Mean | 73.93** | 1.86** | 45.53** |
| Lambda_Var | 2.98** | 38.61** | 4.16** |
| OC_Mean | 15.08** | 1.36** | 3.64** |
| Pricing_Structure | 1.63** | 3.50** | 2.22** |
| Variance | 1.48** | 12.89** | 3.99** |
| OC_Variance | 0.14** | 0.59** | 0.62** |
| Mean × Lambda_Var | 0.35** | 0.07 | 0.20** |
| Lambda_Var × OC_Mean | 0.08** | 0.04 | 0.02 |
| Lambda_Var × Pricing_Structure | 0.05** | 0.62** | 0.44** |
| Lambda_Var × Variance | 0.11** | 0.87** | 0.69** |
| Lambda_Var × OC_Variance | 0.01 | 0.01 | 0.00 |
| Mean × OC_Mean | 1.37** | 0.01 | 0.06 |
| Mean × Pricing_Structure | 0.12** | 1.05** | 11.62** |
| Mean × Variance | 0.28** | 0.46** | 0.55** |
| Mean × OC_Variance | 0.10** | 0.27* | 0.38** |
| OC_Mean× Pricing_Structure | 0.31** | 1.27** | 7.02** |
| OC_Mean× Variance | 0.04** | 0.23* | 0.03 |
| OC_Mean × OC_Variance | 0.12** | 0.55** | 0.00 |
| Pricing_Structure × Variance | 0.23** | 0.39 | 1.71** |
| Pricing_Structure × OC_Variance | 0.05** | 0.53* | 0.08 |
| Variance × OC_Variance | 0.06** | 0.23* | 0.00 |

Note:

* $p < 0.05$

** $p < 0.01$

number of users to participate, irrespective of whether the level of overconfidence is high or low, as shown on the left-hand side of Fig 7(B).

The results presented in Fig 8 demonstrate that under the similar situation (i.e., when the variance of lambda is small, the variance of preference is large) but the mean preference is high, the results show that the two-part tariff try attracting more users than the menu of three-part tariff. In this scenario, even though consumers essentially have a high preference for the service, their demands vary considerably. If the service provider sets the tariff too high, there will essentially be an insufficient number of consumers subscribing to the service, thereby impacting the profit. In other words, the service provider needs to make a trade-off. By lowering the tariff, they can attract more users but will need to sacrifice earnings from consumers with very high demand. Fig 8(B) shows that if consumers have similar price sensitivity (i.e., with a small variance of price sensitivity), it is possible to implement a two-part tariff with slightly lower rates to encourage a larger number of user participation. However, this strategy becomes less effective when the variance of price sensitivity (λ) is larger.

On the other hand, when the more flexible three-part tariff is used, an allowance component in the tariff structure can better encourage consumer participation and profit extraction even in situations with significant heterogeneity among consumers. Thus, regardless of whether price sensitivity is high or low (i.e., irrespective of the degree of heterogeneity in consumers' demand), the number of participants remains more stable.

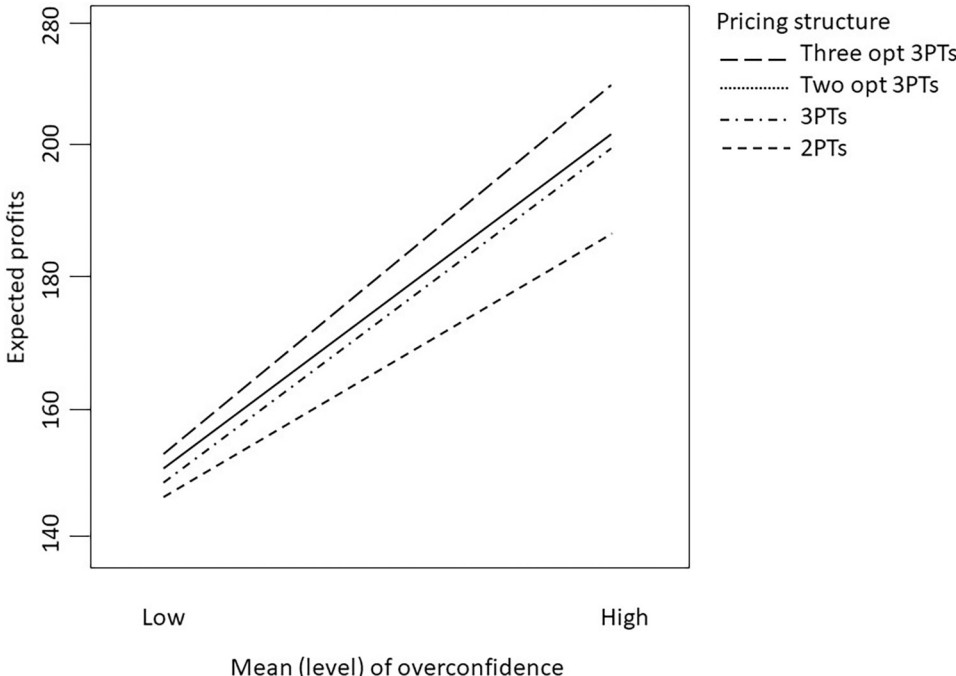

**Fig 3. Effect of overconfidence level on the profitability.**

**Analysis of usage quantity.** Table 7 shows that the mean of consumer preferences has a significant impact on the total usage quantity. Additionally, our numerical analysis reveals that the total usage quantity of consumers can also be influenced by the interaction between mean preferences and the overconfidence level, as well as the optimal tariff structures employed by the service providers. Specifically, the fixed fee and usage price can play a role in determining the usage quantity of consumers.

For a better understanding of the interactions between the overconfidence level, consumer preferences, and tariff structures on usage quantity, we can refer to Fig 9. In particular, Fig 9 (B) shows that when consumers are highly overconfident and have high preferences, a two-

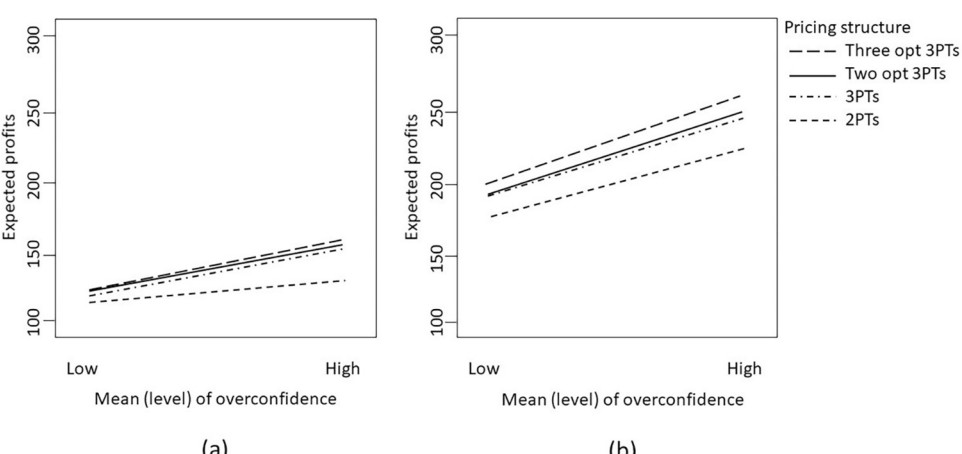

**Fig 4.** Interaction between consumer's overconfidence level and pricing structure on the expected profit when the mean of preference is (a) low; or (b) high.

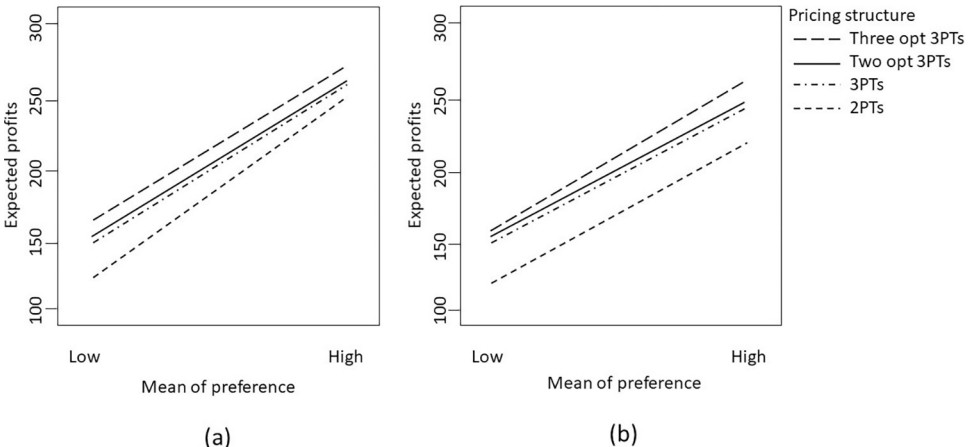

**Fig 5.** Interaction between mean of preference and pricing structure on the expected profit when overconfidence level is high and the variance of preference is (a) small; or (b) large.

part tariff with a high fixed fee and a high usage price will be implemented. Although the associated high fees will lead to a lower number of participants, a service provider can earn a high margin or unit profit from consumers, thus leading to higher overall profit. This is because highly overconfident consumers may be more likely to overestimate their future usage quantity and therefore will be induced to participate by a two-part tariff with a high fixed fee and high usage price (even though their actual consumption is much lower than what they expected).

In contrast, a flexible three-part tariff can take a wider range of consumers into consideration. For example, an appropriate allowance associated with the fixed fee can induce consumers with lower demand to participate while charging for additional units based on a usage price can induce high demand consumers. When facing highly overconfident consumers, the service provider will increase the fixed fee and at the same time lower the allowance. The service provider might also use a similar or even higher usage fee for each additional unit consumed. Since this tariff design will attract more participants, the overall usage quantity will increase as well.

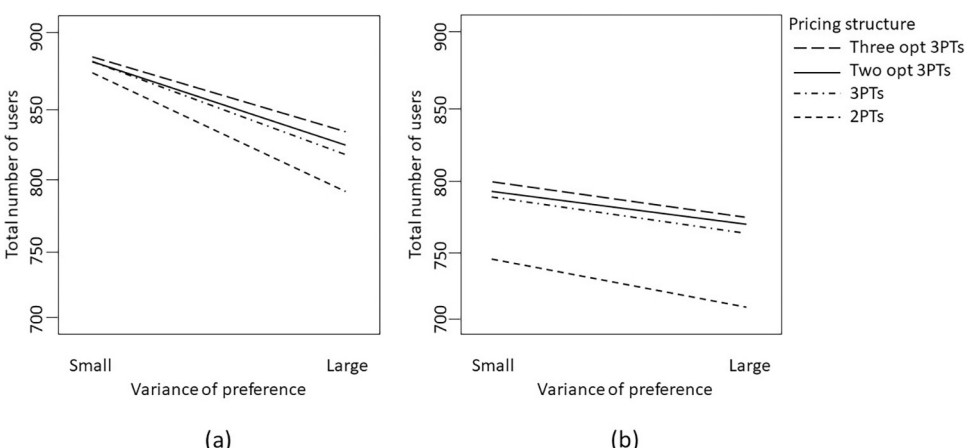

**Fig 6.** Interaction between variance of preference and pricing structure on the total number of user when the variance of price sensitivity (i.e., lambda) is (a) small; or (b) large.

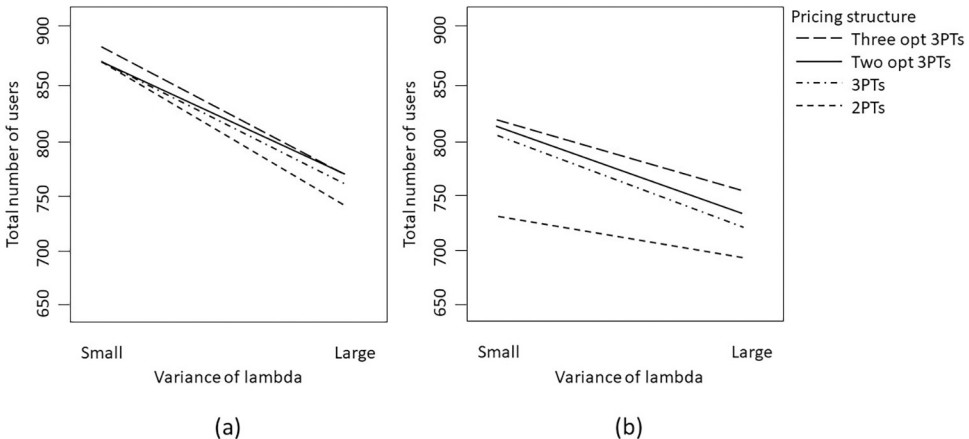

**Fig 7.** Interaction between variance of lambda and pricing structure on the total number of user when the mean preference is low and the variance of preference is (a) small; or (b) large.

Furthermore, when consumers have low preferences (i.e., lower demand) and are less overconfident, a two-part tariff structure with a lower fixed fee and a higher usage price will be implemented. By reducing the fixed fee, this tariff design can lead to a higher number of participants and a higher usage quantity, but the overall profit that can be extracted from the consumers will decrease, as shown on the left-hand side of Fig 9(A). However, when consumers have lower preferences but are highly overconfident, a two-part tariff structure with a higher fixed fee (and a higher usage price) will be implemented, thus leading to a lower usage quantity. Just as we can observe on the right-hand side of Fig 9(A), the usage quantities for different tariff structures under this situation are quite similar.

## Conclusion

In this study, overconfidence behavior may explain why three-part tariff is the optimal pricing strategy for service providers. Consumers overestimate their future consumption of products or services, or their capacity to consume them, which leads to such bias. The cause may be the consumers' inconsistent consumption patterns or marketing from service providers.

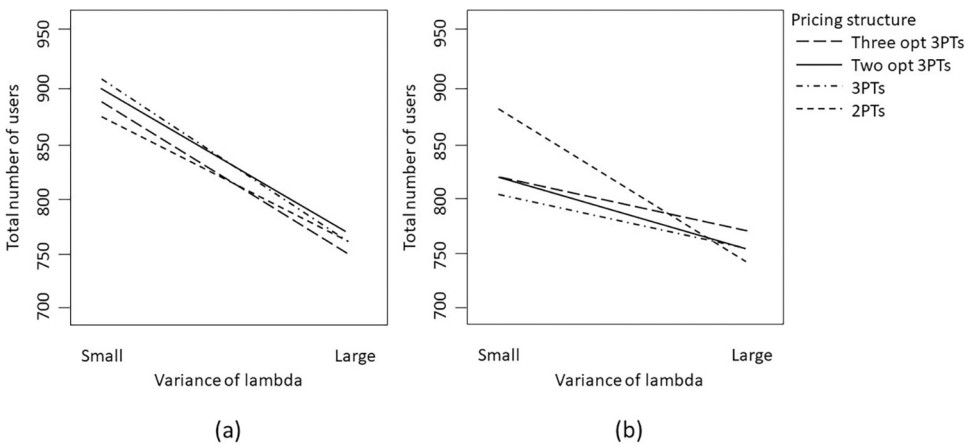

**Fig 8.** Interaction between variance of lambda and pricing structure on the total number of user when the mean preference is high and the variance of preference is (a) small; or (b) large.

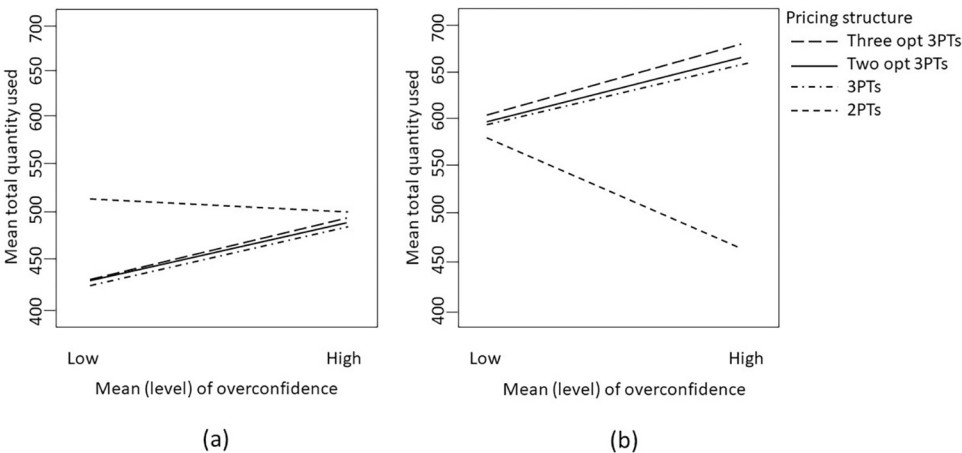

**Fig 9.** Interaction between mean (level) of overconfidence and pricing structure on the expected total usage quantity when the mean of preference is (a) low; or (b) high.

Consumers in the service business, particularly those in this study, frequently overestimate the amount of services used. Under the demand uncertainty generated by overwhelming consumer overconfidence, Grubb [1] describes the benefits of three-part tariffs, particularly when consumers are too accurate for future demand, and suggests that the service provider offer a three-part tariff pricing structure.

Thus, we model overconfidence in the context of overestimating future demand due to consumers' overestimation of service usage or consumption. By creating an analytical model and executing a numerical analysis that uses a genetic algorithm approach, we examine the design of a three-part tariff pricing plan when consumers are overconfident. According to Li et al. [12], if the overconfidence level is too high, then to establish a pretty high fixed fee with a suitably high allowance becomes conceivable. However, if the overconfidence level is too low, the monopolistic service provider can charge low rates with smaller allowances while maintaining a high unit price. In this situation, the service provider may use marketing strategies such as advertisements and promotions to increase customer trust, and thereby the profits. The findings also reveal that as the overconfidence level rises, profits also marginally increase.

This research reveals several main factors and their interactions that influence the profitability, user participation, and quantity of usage of various pricing structures. Several ideas are also provided for when to utilize different pricing structures (under different consumer distributions or market circumstances). For example, we demonstrate that having a menu with multiple three-part tariff pricing schemes can only significantly enhance profits when compared with a two-part tariff scheme, especially at low overconfidence levels and low preferences.

We acknowledge that the results and findings of our numerical analysis could potentially be affected by the specific parameter settings chosen for the analysis. To address this concern, we conducted a sensitivity analysis in our simulation study, employing varied ranges for the parameter distributions. Specifically, we adjusted settings related to the parameters (i.e., distributions used in the simulation experiments) and introduced variations of around 25% from their original values. Our conclusion reveals that these adjustments did not yield significant deviations in the outcomes. This suggests the robustness of our findings against parameter changes, strengthening the reliability of our study and mitigating the likelihood of findings arising solely from the model's construction or parameter settings.

While this research presents the clear benefit of providing useful information into the design and implementation of three-part pricing systems when customers are overconfident,

certain drawbacks are encountered. This study's focus on profit optimization rather than on ethical implications is one of its limitations. However, this study has methodological implications that may be applied to similar systems or situations, in which the manager's focus is different. In the future, this methodology may be extended to consider additional factors, such as enhancing consumer wellbeing, increasing the total usage consumption, or boosting the level of user participation, even when the consumers are overconfident. In this case, we might be able to better understand how to increase the benefit of society when consumer cognitive bias is taken into consideration. Even though we are unable to accomplish it at this time, this study provides directions to others and may serve as a basis for the modeling framework. In addition, this study succeeded in identifying the impacts of the changes in some of the parameters on the optimal solution, however, investigating the nonlinear or higher-order effects of the parameters on optimal solution should be an important direction for the future research.

Furthermore, this study derived from previous works on three-part tariff pricing. We assume that the consumers uniformly distributed within a closed interval have the same level of overconfidence. A highly flexible demand function with different levels of overconfidence will be beneficial and must be considered in future studies. We also view overconfidence as a type of irrational attribute to investigate the psychological aspects of behavior. Future studies can expand the research scope by considering other psychological behavior components that may have a strong impact on how three-part tariff pricing is implemented. Despite the fact that we already have procedures to make sure that genetic algorithms can achieve nearly optimal solution, it may still be possible to investigate more efficient ways of solving optimal solutions for this kind of problem in the future.

## Supporting information

**S1 Appendix. Profit maximization model.**
(PDF)

**S1 File. Data for numerical analysis.**
(XLSX)

## Author Contributions

**Conceptualization:** Shi-Woei Lin, Ruey Huei Yeh.

**Data curation:** Silvia Merdikawati.

**Formal analysis:** Silvia Merdikawati, Shi-Woei Lin.

**Funding acquisition:** Shi-Woei Lin.

**Methodology:** Silvia Merdikawati, Shi-Woei Lin.

**Project administration:** Shi-Woei Lin.

**Resources:** Shi-Woei Lin.

**Software:** Silvia Merdikawati.

**Validation:** Shi-Woei Lin, Ruey Huei Yeh.

**Visualization:** Silvia Merdikawati.

**Writing – original draft:** Silvia Merdikawati, Shi-Woei Lin.

**Writing – review & editing:** Silvia Merdikawati, Shi-Woei Lin, Ruey Huei Yeh.

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
