## [Decision Letter · Decision Letter 0]

14 Oct 2022

PONE-D-22-14682Optimal three-part tariff pricing and marketing strategies for consumer overconfidencePLOS ONE

Dear Dr. Lin,

Thank you for submitting your manuscript to PLOS ONE. After careful consideration, we feel that it has merit but does not fully meet PLOS ONE’s publication criteria as it currently stands. Therefore, we invite you to submit a revised version of the manuscript that addresses the points raised during the review process.

Note that this is a very risky resubmission. One reviewer recommends a major revision and the other rejection, with very solid arguments. Therefore you need to resubmit a significantly improved manuscript that carefully addresses all comments. I believe that you can do it if the model is carefully reformulated, making it elegant and simpler, following the reviewers recommendations. You need to explain and justify your assumptions and evaluate the sensitivity and limitations of your results accordingly. As it stands, you are overconcluding while your results are very weak and easily questionable.  In addition, I want you to consider the ethical implications of your work. How can the market use your results to improve consumer well-being instead of simply exploiting a cognitive bias for profit? Please discuss this in depth.

We look forward to receiving your revised manuscript.

Kind regards,

Carlos Andres Trujillo, PhD

Academic Editor

PLOS ONE

“This study was partially supported by the Ministry of Science and Technology of Taiwan under the grant number MOST 106-2410-H-011-004-MY3 and MOST 109- 2410-H-011-014. Any opinions, findings, and conclusions or recommendations expressed herein are those of the authors and do not necessarily reflect the views of the sponsors.”

“S.W. Lin

Grant Number: MOST110-2621-M-011-001

Grant Number: MOST106-2410-H-011-004-MY3

Funder: Ministry of Science and Technology, Taiwan

http://www.most.gov.tw

Reviewers' comments:

Reviewer's Responses to Questions

**Comments to the Author**

1. Is the manuscript technically sound, and do the data support the conclusions?

Reviewer #1: Yes

Reviewer #2: Partly

2. Has the statistical analysis been performed appropriately and rigorously? 

Reviewer #1: Yes

Reviewer #2: I Don't Know

3. Have the authors made all data underlying the findings in their manuscript fully available?

Reviewer #1: Yes

Reviewer #2: Yes

4. Is the manuscript presented in an intelligible fashion and written in standard English?

Reviewer #1: Yes

Reviewer #2: Yes

5. Review Comments to the Author

Reviewer #1: This paper studies the benefits of three-part tariffs under demand uncertainty caused by overconfident consumers. The research question is interesting and results are reasonable and potentially useful in practice. I have several comments that may be useful for the authors to strengthen this paper’s exposition and contribution.

1. The demand function, especially the uncertainty component, should be clearly explained.

2. How the overconfident consumers behave should be justified. What is the impact on the firm’s profit?

3. Some mathematical formulas are too lengthy or tedious, without transition or comments, for example equations (15), (25) and others.

4. The comparative statics seem interesting. Can you derive similar analytical results, in addition to numerical study?

5. Part 4 comes too late, and I suggest to discuss consumer overconfidence at the beginning.

I recommend a major revision.

Reviewer #2: Review of the paper optimal tariff pricing and marketing strategies for consumer overconfidence

Summary

This paper attempts to understand how firms may determine the optimal three-part tariff structure when facing overconfident consumers. Overconfident consumers tend to miscalculate their future consumption which may be used by firms to maximize profits. Firms may charge a higher fix cost for a specific number of unit allowance and a small usage fee for units consumed above that specified number, expecting overconfident consumers to pay the fix cost, and not consuming all the units’ allowance. In this context, the paper looks for the optimal parameters and the optimal menu of three parts tariff maximizing firms’ profits.

Main comments

• My first comment must be about how the paper has been written. The paper is very confusing to read, some parameters of the model are not well explained (for instance, the role of \\theta_1 and \\theta_2), and there seems to be a changed of variable at some point about the consumer types (being \\theta at the beginning of section 2 and then becoming \\theta+r in my interpretation). Many other typos of this type are found along the paper, but I will not mention all of them. I think is the role the authors to provide a paper more carefully written making the role of the reviewers easier.

• Other important issue to mention is that the paper does not really focus on the optimal three-part tariff structure because their solution depends on several specific parameters and assumptions. In the best-case scenario, they can find some local optimums but not the global optimum of the problem. This is even the case with the algorithm approach. This concern it's more problematic given the nonlinear nature of the profit functions they have.

• These nonlinearities are also relevant for comparative static analysis where the authors use only two different values for the marginal cost and the consumer price sensitivity. The effect found cannot be generalize to a different range of numbers and the authors do not explain why the numbers selected are the most relevant in this context. In particular, the price sensitivity equal 1.

• I am aware that the difficulties to solve these types of models open the door to this type analytical and numerical approaches, but they are usually a tool used to understand the role of a particular mechanism. However, the mechanism in your paper was already known and your contribution is to find the optimum three-part tariff. In my opinion, this methodology does not allow you to solve this problem.

• Respect to the assumptions of the model, I wonder why the authors assume that the level of overconfidence is the same for all consumers in the population. Moreover, they assume that there is no relationship between their types and their level of overconfidence. Of course, these assumptions help to maintain the uniform distribution of types and the tractability of the model, but it also limits its applications. Could the authors extend their analysis to other types of distributions?

• At the beginning, the model assumes that consumer types are uniformly distributed on the range between zero and one. However, in the profit maximization of the firm the consumers are distributed between \\theta star and \\theta upper bar. Similarly, in the numerical analysis consumer types are not between zero and one. The uniform distribution between zero and one make it easier to calculate the consumer surplus but then you cannot change the range of the distribution. the authors should explain better the reasons behind this change.

Minor comments

• Respect to the analysis of the number three-part tariffs in the menu offered by the firm, it is not surprising that you find dot a menu with more options dominates the others. Basically, more pricing options allow you for a better discrimination of the consumers and therefore higher profit. Considering this a more interesting analysis will be to determine when this discrimination capacity matters the most.

• The authors use the demand function coming from Tirole (1995) and they argue that they're reasonable assumptions for the appropriateness of this demand function in terms of utility theory. I think the authors should discuss those assumptions in the paper and to discuss how appropriate is this function to study overconfidence.

• More explanation is needed in the genetic algorithm. What determines the selection of individuals in each iteration? Do you use any measure of fitness in this case, or everything is random from the original population distribution?

• It is important to differentiate throughout the paper between the consumer surplus and expected consumer surplus, based on the difference between consumption and expected consumption that the authors do not make.

• In section 2.1, you include a double characterization of the consumers. First, you mentioned that there are two types of individuals, \\theta_1 and \\theta_2, but then you use these two types to characterize regions of consumers. The explanation is confusing, and you should differentiate the role of these categories better.

• The last paragraph section 2.2 we shall have \\theta+r in many places, it looks like a mistake. But an alternative explanation is that they are using \\theta as \\theta+r which is confusing giving his presentation of the model in Section 2.

• In the parametrization of the model, which are reasonable levels of overconfidence a why?

• Table 2 and figure 2 show the same information they should move one of them to the appendix.

• A sensibility analysis is needed in the simulation respect to the main parameters to determine in what range your conclusions are valid.

• In section 2.3, the firm should not maximize respect to marginal costs that's something that you should take as given.

To conclude, I think the paper fails to find the optimal three-part tariff structure with overconfident consumers given my comments above. For this reason, I suggest rejecting the paper.

6. PLOS authors have the option to publish the peer review history of their article (what does this mean?). If published, this will include your full peer review and any attached files.

Reviewer #1: No

Reviewer #2: No

---

## [Author Response · Author response to Decision Letter 0]

31 Mar 2023

We carefully considered the comments offered by the academic editor and two reviewers and made corresponding changes. In a separate document (i.e., a 22-page document included in the generated pdf file), we provide detailed responses to the comments and explain in detail how we revised the paper based on those comments and recommendations.

---

## [Decision Letter · Decision Letter 1]

23 Jun 2023

PONE-D-22-14682R1Optimal three-part tariff pricing and marketing strategies for consumer overconfidencePLOS ONE

Dear Dr. Lin,

Thank you for submitting your manuscript to PLOS ONE. After careful consideration, we feel that it has merit but does not fully meet PLOS ONE’s publication criteria as it currently stands. Therefore, we invite you to submit a revised version of the manuscript that addresses the points raised during the review process.

 As you can see, one reviewer recommends accepting the paper but the other rejecting it. I think you have done a remarkable job improving the paper, but the remaining concerns must be addressed. Reviewer 2 offers very precise feedback on what is still missing. I want to highlight the importance of showing that your results are not a trivial artifact of the model construction, but a true insight. That will be the key for me to make a final decision in the next round. The other comments are very doable. 

We look forward to receiving your revised manuscript.

Kind regards,

Carlos Andres Trujillo, PhD

Academic Editor

PLOS ONE

Reviewers' comments:

Reviewer's Responses to Questions

**Comments to the Author**

1. If the authors have adequately addressed your comments raised in a previous round of review and you feel that this manuscript is now acceptable for publication, you may indicate that here to bypass the “Comments to the Author” section, enter your conflict of interest statement in the “Confidential to Editor” section, and submit your "Accept" recommendation.

Reviewer #1: All comments have been addressed

Reviewer #2: (No Response)

2. Is the manuscript technically sound, and do the data support the conclusions?

Reviewer #1: Yes

Reviewer #2: Partly

3. Has the statistical analysis been performed appropriately and rigorously? 

Reviewer #1: Yes

Reviewer #2: Yes

4. Have the authors made all data underlying the findings in their manuscript fully available?

Reviewer #1: Yes

Reviewer #2: Yes

5. Is the manuscript presented in an intelligible fashion and written in standard English?

Reviewer #1: Yes

Reviewer #2: Yes

6. Review Comments to the Author

Reviewer #1: In the previous round, I provided 5 comments on the modeling, assumption, technical results and writing. In this round, the authors addressed all my comments very satisfactorily. The detailed changes are well documented in the response letter. Therefore, I have no additional comment and am happy to recommend to accept this manuscript.

Reviewer #2: Thank you for your attention to several of my comments in the previous round. You have improved the clarity of the paper in many of the sections, but there are still different issues that could be better explained. In this revised manuscript, the introduction and numerical analysis are much clearer, but the model needs more work. In any case, my main concern now is with the contribution of the paper. My comments to this version of the paper are the following:

1. Contribution: Citing the authors: “The main contribution of our study differs from that of previous studies, which paid attention mainly on the optimal three-part tariff. In our study, we not only consider the cognitive bias in the optimal three-part tariff but also investigate the three-part tariff pricing menu.” In the paper, the authors showed that a three-part tariff menu with more tariffs plans dominate one when less tariff plans for all the different parameters on their numerical exercise comparing menus with three, two and one tariff. As I mentioned in my previous review in comment 7, this is not a surprising result because more tariff plans allow the firm to discriminate better. A higher ability to discriminate among clients helps the firm to increase its profits. Moreover, the results of the experiment in section 5 shows that the menu is more profitable and increase participation the smaller the general over-confidence level. At a first glance, this result seems really interesting, but my intuition is it comes actually from the model construction. A lower over-confidence level implies we have two qualitatively different type of clients as shown in panel b and c in Figure 1. The clients in panel d are qualitatively equal to those on panel c. As a consequence, the advantages of the discrimination possibilities imply higher profits and user participation as the more qualitatively different are the clients of the firm. Then, the novelty of the contribution of the paper is not very convincing for me and I do not know if it is enough to grant a publication in the PLOS One journal.

2. Model: Section 2.1 needs to be fully rewritten. The explanation of the consumer’s type is correct but very confusing. First, the WTP equations 3 and 4 are really the consumer surplus not the willingness to pay. Second, I think the consumer indexes add confusion instead to clarify the analysis. Third, the figure 1 need to be more clearly explained and connected to the model and the intuition of the model. Fourth, the indifferent consumer \\theta* could be higher than \\theta_2 as defined by table 1 and smaller than 1; as a consequence, you can have a third case that I think is necessary to analyze. Respect to section 2.2, I will move many of the calculations to the appendix. Also, I think you need to change the positions of section 3 and 4.

3. Algorithm: The explanation of the algorithm has improved a lot, but I think the authors should connect it better with the model. For instance, it is not clear to me is the chromosomes are the thetas in the model and how those evolve and is related with the distribution of the types in the population.

4. Parametrization: The recognition of the limitations of the numerical analysis is necessary, but I still think that a discussion on how sensitive the results to the parameters of the model are is in place.

To conclude, given my comments above and in particular the lack of novelty of the contribution in the paper since my point of view, my recommendation is to reject.

7. PLOS authors have the option to publish the peer review history of their article (what does this mean?). If published, this will include your full peer review and any attached files.

Reviewer #1: No

Reviewer #2: No

---

## [Author Response · Author response to Decision Letter 1]

28 Aug 2023

We carefully considered the comments offered by the academic editor and two reviewers and made corresponding changes. In a separate document (i.e., a 21-page document included in the generated pdf file), we provide detailed responses to the comments and explain in detail how we revised the paper based on those comments and recommendations.

---

## [Decision Letter · Decision Letter 2]

15 Jan 2024

Optimal three-part tariff pricing and marketing strategies for consumer overconfidence

PONE-D-22-14682R2

Dear Dr. Lin,

We’re pleased to inform you that your manuscript has been judged scientifically suitable for publication and will be formally accepted for publication once it meets all outstanding technical requirements. I know it was a challenging process but I hope it was constructive.

Kind regards,

Carlos Andres Trujillo, PhD

Academic Editor

PLOS ONE

Additional Editor Comments (optional):

Reviewers' comments:

Reviewer's Responses to Questions

**Comments to the Author**

1. If the authors have adequately addressed your comments raised in a previous round of review and you feel that this manuscript is now acceptable for publication, you may indicate that here to bypass the “Comments to the Author” section, enter your conflict of interest statement in the “Confidential to Editor” section, and submit your "Accept" recommendation.

Reviewer #2: All comments have been addressed

2. Is the manuscript technically sound, and do the data support the conclusions?

Reviewer #2: Yes

3. Has the statistical analysis been performed appropriately and rigorously? 

Reviewer #2: Yes

4. Have the authors made all data underlying the findings in their manuscript fully available?

Reviewer #2: Yes

5. Is the manuscript presented in an intelligible fashion and written in standard English?

Reviewer #2: Yes

6. Review Comments to the Author

Reviewer #2: Most of my comments have been addressed by the authors. We have some differences on the use of certain definitions but I consider them minor points. I think the new version is much more clear and the introduction of two part-tariff helps to have a more clear contribution and a more interesting message for managers and researchers in this topic.

7. PLOS authors have the option to publish the peer review history of their article (what does this mean?). If published, this will include your full peer review and any attached files.

Reviewer #2: **Yes: **Miguel Angel Martinez-Carrasco

---

## [Editor Report · Acceptance letter]

26 Jul 2024

PONE-D-22-14682R2 

PLOS ONE

Dear Dr. Lin, 

I'm pleased to inform you that your manuscript has been deemed suitable for publication in PLOS ONE. Congratulations! Your manuscript is now being handed over to our production team.

Kind regards, 

on behalf of

Dr. Carlos Andres Trujillo 

Academic Editor

PLOS ONE